# The Human Superficial Fascia: A Narrative Review

**DOI:** 10.3390/ijms26031289

**Published:** 2025-02-03

**Authors:** Caterina Fede, Claudia Clair, Carmelo Pirri, Lucia Petrelli, Xiaoxiao Zhao, Yunfeng Sun, Veronica Macchi, Carla Stecco

**Affiliations:** Department of Neurosciences, Institute of Human Anatomy, University of Padova, 35121 Padua, Italy; claudia.clair@unipd.it (C.C.); carmelo.pirri@unipd.it (C.P.); lucia.petrelli@unipd.it (L.P.); xiaoxiao.zhao@studenti.unipd.it (X.Z.); yunfeng.sun@studenti.unipd.it (Y.S.); veronica.macchi@unipd.it (V.M.); carla.stecco@unipd.it (C.S.)

**Keywords:** fascia, superficial fascia, hypodermis, matrix, autonomic innervation, blood vessels, lymphatic

## Abstract

In recent years, the interest in the comprehension of the fasciae has significantly grown, together with the necessity of finding a consensus for a terminology of the fasciae in the research and clinical fields. Furthermore, it is becoming necessary to categorize the various types of fascia (superficial, deep, visceral, neural) since they possess different anatomical characteristics, and are implicated in different pathophysiological pathways. While in the past we have described the deep/muscular fascia, the aim of this work is to summarize and catalog the information relating to the human superficial fascia (thickness, cellular end extracellular matrix component, innervation, vascularization).

## 1. Introduction

Anatomists, academics, and professionals in different health-oriented fields, increasingly use the term fascia but there is still the absence of a clear consensus on the definition and significance of this term, thus making it impossible to compare results and descriptions among different authors or papers [1,2,3,4]. Moreover, most of the authors are focused on the comprehension of the deep/muscular fascia, in its role in force transmission and proprioception, and in its implication in movement and muscular force, together with its alterations in pathological processes. However, the superficial fascia (SF) in recent years has been recognized as a key element that organizes the hypodermis. It is a thin fibrous layer mainly formed by collagen and elastic fibers, but also, containing fat lobules, vessels, and nerves [5,6,7], and in some regions including also muscle fibers [6]. 

The superficial fasciae (SFs) that were first discovered and studied are those of Scarpa (1801), at the level of the abdomen, and of Colles (1819), at the level of the pelvic and inguinal area [8]. For a long time, it was thought that the SF was present only at the level of these districts but more recently it has been recognized that it runs across the entire surface of the body [9,10,11,12,13,14,15], together with the dermis, presenting differences based on the location and function of the area it covers. 

Anatomically, the SF separates two parts of the hypodermis: the superficial adipose tissue (SAT) and deep adipose tissue (DAT) [16,17,18] (Figure 1A). These two layers present different features and functions: the SAT is strongly connected with the skin and shares with it the innervation, probably collaborating in exteroception, whereas the DAT is a less organized layer, rich in hyaluronan and water with few adipose cells and collagen fibers, with the main function of separating both physically and functionally the SF from the deep one, allowing them to respond independently to different external and internal stimuli [19]. Passing between these two layers, the SAT and the DAT, and dividing them, the SF permits the separation of the skin from the musculoskeletal system, allowing an independent and normal sliding of these two components. Moreover, the SF splits to surround vessels and nerves, ensuring their patency, and forms the so-called retinacula cutis, which has the function of connecting the SF to the skin (retinacula cutis superficialis, or skin ligaments) and to the deep fascia (retinacula cutis profundis), to provide a dynamic and flexible anchor of the skin to underlying tissues [16,20]. The deep septa are rare, thin, and oblique and have an apparent lesser organization, allowing great autonomy between superficial and deep fascia. In contrast, the superficial septa are short, vertically oriented, dense, and connect the SF to the skin with a different orientation depending on the region [16,21,22]. The presence of retinacula in the SAT determines the formation of polygonal-oval lobes of fat cells with a mean circularity factor of 0.856 (±0.113) to 1, while in the DAT retinacula define large, flat, and polygonal lobes of fat cells (circularity factor: mean 0.473 to 1, SD: 0.07) [23].

At the level of the sole of the foot and the palm of the hand, the SF is apparently absent because it fuses with the underlying deep fascia, forming the so-called palmar fascia and plantar fascia [16]. Even at the facial level, points of adhesion between the superficial and deep fascia have been identified, over either the masseter or buccinator muscles, with the function of separating adjacent adipose compartments, or adipose compartments and anatomical spaces [24]. Moreover, in the face, the SF envelopes all the mimical muscles, and it is also called the SMAS (superficial musculo-aponeurotic system) of the face. In the skull, it becomes very fibrotic, and it is called galea capitis. In the gluteal region, the SF was clearly observed in the upper region, but becomes more difficult to describe in the lower gluteal region, which appears to be made up entirely of fat lobules with honeycomb septal walls [18].

The SF presents the typical characteristics of fibrous connective tissue formed by cells and a fibrous and aqueous extracellular matrix (Figure 1B): the loosely packed interwoven collagen fibers are mixed with a high content of elastic fibers. Moreover, it is populated by immune cells, and pervaded by a thin and huge network of blood and lymphatic vessels. Each component of the tissue defines specific functional characteristics of the fascia, which are described below and further explained.

## 2. Results

### 2.1. Thickness of the Superficial Fascia

The thickness of the SF can vary depending on its anatomical location, the individual’s body mass index (BMI), age, gender, and physiopathological condition. By ultrasound imaging, the SF appears as a hyperechoic tissue that presents a good contrast with the surrounding tissues [7,25], but the variability caused by the instrument, the probe and patient position, and intra- and interpersonal variation has to be taken into account. As demonstrated by Hammoudeh et al. and Pirri et al. [7,26], different regions of the body show marked differences in the thickness of the SF and in the number of layers it is composed of, appearing linear, laminate, bi-laminate, or tri-laminate. Moreover, the thickness of the subcutaneous tissue is significantly and positively correlated with the total thickness of the SF and the mean thickness of the various layers that form it [26]. In general, in the posterior region of the body, the SF appears less defined, but thicker, especially in the posterior part of the trunk, thigh, and arm [27]. It is also thicker in the region of the trunk and thighs, compared to the peripheral regions of the body [6]. Gender differences are also observable: Abu-Hijleh observed major thicknesses at the level of the same areas in females with respect to males [28], whereas other authors analyzed that the thickness is greater in the lower abdomen of the male (mean 528.336 ± 38.48 µm) than that in females (mean 390.822 ± 36.24 µm) [29,30].

Moreover, a wide discrepancy between the thicknesses of the fasciae measured according to histological sections and ultrasound (US) images was noted, probably due to tissue dehydration and shrinkage during the histological protocols. This evidence highlights the difficulty of comparing studies carried out with different protocols and techniques. In any case, the patterns measured by US and histology were similar, with a correlation (r = 0.918) highly significant (*p* < 0.01) [7]. In detail, Pirri et al. have demonstrated that, at thigh level, the average thickness of the SF measured by US imaging is equal to 490 ± 140 µm (anterior portion), 520 ± 100 µm (medially), 420 ± 120 µm (laterally) and 500 ± 110 µm (posterior portion). The measurements performed by histology give a mean thickness of 146.6 ± 31.5 µm, but it does not appear uniform: the mean thicknesses were 153.2 ± 39.3 µm anteriorly, 128.4 ± 24.7 µm medially, 154.0 ± 28.8 µm laterally, 148.8 ± 33.2 µm posteriorly [25].

The dorsal trunk has the thickest SF with a mean thickness of 600 µm (0.6 to 0.7 mm), followed by the lumbar region (similar thickness, but slightly thinner on average, ~580 µm) [26].

The anterior regions appear thinner, with an average SF thickness of 560 µm in the thorax [26]. At the level of the abdomen, the SF is multilayered (3 to 7) in the midline, with a reduction in the number of layers as one proceeds towards the lateral portion, due to the progressive fusion between them. The thickness of the SF is greater in the lower portion of the abdomen than in the upper one, with a mean thickness, respectively, of 528.336 ± SE 38.48 µm and 364.165 ± SE 22.49 µm in the male; 390.822 ± SE 36.21 µm and 315.822 ± SE 56.93 µm in the female [30].

At the nipple-areolar complex (NAC) there is a hyperdense line of SF, with an average thickness of 309 ± 171 µm, visible immediately below the skin in the sagittal sections [31]. This line turned under the NAC and at this point, the line became very thick. This measurement is important for infra-alveolar mammoplasty [32]. However, the thickness of the mammary fascia tends to change in different stages of life, especially in women [33].

The SF of the upper limb shows a mean thickness of 450 ± 100 µm and is thicker in the posterior region of the arm (530 ± 100 µm) than in the anterior one (400 ± 100 µm), while no statistically different values were noted between the anterior (400 µm) and posterior (410 µm) regions of the forearm [27].

All the main mean values of SF evaluated by US imaging are summarized in Table 1.

### 2.2. The Cellular Population

Like in the deep fascia [34], the predominant cell population of the SF tissue is fibroblasts [35], randomly distributed, with the function of maintaining the structural integrity of the tissue, and of producing precursors of the extracellular matrix, such as collagen and elastic fibers, contributing to the organization and remodeling of the matrix. Numerous adipocytes are found in the midst of the collagen fibers, in small clusters. Some studies demonstrated that some preadipocytes are found in the SF, which gives origin to the retinacula cutis in superficial and deeper adipose layers [36].

Moreover, a density of 20.4 ± 9.4/mm^2^ of mast cells was observed in the SF of the subumbilical area, between the collagen fibers, near the wall of the blood vessels, and near the nerve fibers [37]. These cells are able to intervene together with other inflammatory transient cells, in early inflammatory stages, in the tissue healing and regeneration process, confirming their involvement in these processes in the subcutaneous tissue and the SF [38].

Moreover, the SF, having ancestral origin from the panniculus carnosus present in non-human mammals, which serves to produce local movement of the skin [16], can present in some regions, muscle fibers, such as in the neck (indicated as the platysma muscle), in the face (the SMAS or superficial muscular aponeurotic system) [39,40], in the anal region (external anal sphincter), and in the scrotum (the dartos fascia) [5,6]. Really, in all the regions, muscular fibers were found inside the SF.

### 2.3. Fibrous Component of the Superficial Fascia

The SF is composed of fibro-fatty layers interconnected and made of loosely packed collagen fibers (Figure 1B), intertwined with abundant elastic fibers, both vertically and horizontally arranged, crossed by blood vessels and nerves along all three axes [28,41]. The thickness and the number of sub-layers of SF in the different regions of the body are probably proportional to the daily stress experienced in that site, confirming that the specific composition of the tissue can have direct implications for its mechanical properties and functionality [26].

Collagen fibers have great tensile strength: they can withstand considerable tensile forces without a significant increase in their length, but at the same time they are flexible [16].

The SF contains a high amount of elastic fibers forming a 3D network, with a prevalence of fibers on the transversal axis [42]. They form a lacework pattern, in some areas perforating the collagen bundles at different levels, permitting protection of the nerves, and at the same time enabling the proper functioning of blood and lymph vessels. Some authors evaluated that the ratio between collagen and elastic fibers is equal to 1:1 at the level of the lower abdomen; for instance, Pandey and coauthors quantified the collagen fibers as equal to 13.455 ± 3.960% and the elastic fibers as equal to 14.602 ± 5.244% [29]. This percentage of elastic fibers also corresponds with the data published by Pirri et al. [42], who found 13.5% of elastic fibers in the SF of the thigh, ten times more than the fascia lata (deep fascia of the thigh). Really, this amount can physiologically change with aging: in young people the SF is more elastic, allowing the hypodermis to respond adequately to stress stimuli coming from all dimensions without being damaged, and therefore returning to its initial state [16]. Elastic fibers in SF do not form bundles, but they are visible as a web with many branches and anastomose with other fibers. The elastic fibers are thinner than collagen ones and stretch easily with a high capacity to return to their original length [16].

The mechanical tests performed in the thoracic and abdominal regions, observed that the SF is anisotropic, with a more rigid and tenacious nature along the lateral-medial direction, compared to the cranio-caudal one [43]. Furthermore, the thoracic region exhibited significantly greater strength and resultant Young’s modulus compared to the abdomen (with greater results along the latero-medial direction), but the deformation at break was in both regions almost independent of the direction of load, thanks to the high content of elastic fibers. Stress-relaxation tests highlighted the viscous behavior of the SF, with no significant differences in stress decay between different directions: most of the stress reduction occurred in the first minute of rest, reaching on average 37% of residual stress after 300 s [43].

In pathological conditions, such as in the presence of a scar, the load-bearing capacity of the tissue is reduced by 30% and persists for several years after the formation of the scar [44].

### 2.4. Innervation

Although the SF has been reported as the second most innervated tissue after the skin (Fede et al., 2020), its innervation is still little studied. It has been shown that its innervation rate is higher than that of the deep fascia: the skin presented an innervated area equal to 0.73 ± 0.37%, the SF 0.22 ± 0.06%, and the deep one 0.17 ± 0.17% [45]. The nerve fibers in the SF showed a density of 33.0 ± 2.5/cm^2^: some nerve bundles are larger (mean diameter of 21.1 ± 12.2 µm), but the majority of the nerves pervading the tissue are very thin (average diameter of 4.8 ± 2.6 µm), showing a huge network of small nerve fibers supplying the tissue [46]. The nerve fibers are found around the blood vessels, close to the adipocytes, and in the connective tissue itself [46]. The innervation in the midst of the collagen bundles confirms the sensorial role of the SF and its possible implications in nociception, although no Pacinian corpuscles and/or Ruffini corpuscles were found in the abdominal and hip regions, contrary to what has been observed at the level of the plantar fascia [16].

The positivity for the immuno-staining of the tyrosine hydroxylase (TH) marker confirmed the presence of a sympathetic autonomic nervous system in the SF, with a relative percentage of 33.82% and an S100/TH positivity ratio equal to 2.96, suggesting its possible role as vasoconstrictors and regulators of vascularization in the SF [46]. However, some of these sympathetic endings do not terminate on the vessels, so their function is still a matter of debate: they may have a trophic activity [47], a role as pain modulators, mediating mechanical allodynia evoked by touch [48], or may have a role in the control of fascial tone and fascial stiffness [49,50].

The specific innervation of the SF and its strong relationship with the skin suggest that this innervation is part of dermatomeric perception [19]. Moreover, the huge presence of TH-positive fibers permits demonstration that stress, temperature, and trauma may affect sympathetic activity not only in the skin, but also in the SF, with a consequent definition of its function as fundamental for thermoregulation, exteroception, and pain modulation.

### 2.5. Blood Supply and Lymphatic Net of the Superficial Fascia

The SF is richly vascularized by a dense and thin network of blood vessels, composed of arteries, veins, capillaries, and lymphatic segments that “run” inside the SF, both longitudinally and transversally, and which cross it at full thickness, as demonstrated in Scarpa’s fascia. The surface is occupied by arterial vessels (6.20 ± 2.10%) and venous vessels (2.93 ± 1.80%) in the SF of the abdomen [51], but it seems that there are regional differences. Indeed, Tao et al. found at the level of the forearm a smaller quantity of blood vessels, equal to 2.7 ± 2.1% [52]. The vessels present a diameter between 13 and 65 μm, which had an average of 54.24 ± 15.80 µm for arteries, 60.10 ± 1.04 µm for veins, and 17.28 ± 3.84 µm for smaller vessels [51]. All these vessels connect to each other, creating a rete mirabile and extending along straight and curved paths in close contact with the cellular and extracellular components of the fascia matrix. They are well branched with a density of crossing points equal to 3.40 ± 1.90 per mm^2^ and with an optimal homogeneity of spatial distribution, confirming that the entire space of the SF is pervaded by vessels.

Tao and colleagues, analyzing the full-thickness area of the forearm, from skin to muscle, showed that the branches of the intermuscular space and the septal cutaneous arteries join together to form longitudinal arterial chains at the level of the SF, repeatedly ramified [52]. Also, the cadaveric studies of Wang et al. in the lower limb demonstrated that the perforators had three main branches: two cutaneous (vertical and oblique, which enter the subdermal layer) and one SF branch, which travels in the SF and connects with the nearby perforators [53]. The SF network sends branches into adipose lobules to form the capillary networks, permitting the superficial and deep fascial networks to communicate with each other.

Recently, Albertin et al. demonstrated a lymphatic plexus inside the SF. The lymphatic vessels appeared as flattened channels with an average diameter of 19.5 ± 4.77 µm [54]. The lymphatic vessels also head towards the dermis, crossing the SAT and flanking the retinacula cutis, until they reach the vicinity of the hair follicles, sebaceous glands, and sweat glands, forming a coiled tubular structure. This network of lymphatic vessels accompanies adjacent veins and arteries, branching out in different directions, both longitudinally and transversally, following the orientation of the collagen and elastic fibers. Also, Friedman and coauthors observed the lymphatic vessels in the dermis, SF (Scarpa fascia), and loose areolar tissue but not in deep or superficial fat, in the human anterior abdominal wall. They reported that the highest concentration was in the dermis (mean of 82.6%), whereas the Scarpa fascia contained 9.4% of lymphatics [55]. Albertin and coauthors, on the contrary, evaluated the positive area% (with the endothelium positive to factor D2-40) from the ratio between the total immunoreactive area and the total sampled area, showing that in the dermis it was equal to 0.095 ± 0.018%, not significantly different with respect to the SF layer (0.122 ± 0.029%; *p* > 0.05) [54].

Thanks to the injection of dye at the subareolar level and in the glandular tissue of the upper lateral quadrant of the mammary gland, Wuringer et al. demonstrated the presence of lymphatic vessels embedded into the SF also at the level of the breast, showing how some lymphatic vessels isolated within the SF drain toward the axillary lymph nodes [56].

### 2.6. Pathological Implications

The comprehension of the macroscopic and microscopic characteristics of the SF can allow us to better understand the specific alterations that occur in fascial dysfunctions, facilitating a more targeted approach to treatment and therapies. As happened with the deep fascia, which was initially considered only as an inert collagen layer, the SF has long been ignored. Instead, it has its own specific characteristics; it is richly innervated and vascularized, hosts a lymphatic plexus, and consequently can be altered in various pathological situations.

First of all, being strongly connected to the subcutaneous adipose tissue, it is involved in cellulite.

Cellulite is a common aesthetic condition, which affects the majority of women, characterized by the heterogeneous appearance mainly of the skin that covers the gluteal and posterior thigh regions [57]. By studying the diseased tissue in full thickness, it has been seen how the heterogeneity at the skin level is caused by an imbalance between the forces of tissue containment (especially determined by the retinacula cutis) and the forces that push outward (determined by the increase in the volume of adipose cells following an increase in BMI or dysregulation of fluids) [58]. Fibrosis of the subcutaneous connective tissue, shortening the septa and altering the skin surface, is therefore linked to the interaction between containment and extrusion forces, which increase interstitial pressure [59]. The involvement of the SF in this pathological pathway has been proven by the presence of short and oblique septa associated with cellulite dimples, which for 93.4% originate from this structure (the remaining septa originate from the deep fascia) [57] and a thickening of the SF already present at non-severe cellulite levels [60].

Moreover, the SF is thicker in obese people with respect to control ones [23]. The alterations do not seem uniform, but according to Pandey et al. [29] the SF of the lower abdomen has a relatively lower content of elastic support, collagen, and hydroxyproline compared to the upper one, and this could be one of the reasons why in obesity there is a greater sagging fold of the skin in the lower part of the abdomen compared to the upper part.

Regarding the wound healing process, it is proven that the SF is fundamental, intervening by mobilizing its extracellular matrix, acting as a temporary matrix in the wound bed [37]. Once mobilized, the fascial matrix loses its elastic properties and dehydrates becoming a rigid fibrotic scar [61]. However, this reorganization of the tissue, especially when it causes thickening of the fascia and retinacula, triggers inflammation and stiffening of the region adjacent to the scar [62]. Since there are differences in the thickness of the SF, age, and gender-dependent, as already reported, a different involvement of the tissue in the wound healing processes has also been observed: the combination of thinner skin with thicker SF, a common condition in women, can cause an easier violation of the fascial compartment by a skin lesion. This would mobilize the matrix more often in women and inflict larger and more pathological scars. Older patients, on the other hand, have a lower risk of extreme scarring as the SF becomes thinner with age [63].

Knowing the strong interaction between the SF and the lymphatic vessels, it can be postulated that these two factors can influence each other, and in both directions, in cases of problems such as lymphedema. This theory is supported by works which observed a significant increase in the thickness of the SF and retinacula cutis in swollen legs compared to normal limbs. The mean relative increase in the thickness of the SF was 206.0% more than in the control group, while no significant difference was demonstrated between the thickness of the fascia in primary versus secondary lymphedema [64]. Really, these data are not confirmed by Pirri et al. [65], who did not find alterations in SF thickness in the limbs with lymphedema compared with the contralateral side. Really, it is possible that lymphedema leads to alteration of the fascia, and its consequent and eventual damage, but at the same time, the alteration of the SF can compromise the lymphatic drainage present within it, as the collagen and elastic fibers guide the lymphatic flow while also determining a low-resistance path, facilitating the movement of the interstitial fluid [66].

The SF can also be used in reconstructive surgery for the creation of fascial flaps, such as the parascapular fascio-cutaneous free flap or cutaneous flaps [67], thanks to its high vascularization and capacity for adherence to underlying tissues, or it can be employed in cosmetic surgery to lift layers of the lateral midface soft tissues, obtaining significant and long-lasting lateral midface rejuvenation [68].

It can also act as a barrier against tumors that develop at the subcutaneous level: benign lesions in fact remain separated from the fascia, while malignant lesions adapt to its presence even penetrating its margins. These characteristics allow the SF to be used to facilitate the categorization of soft tissue tumors as benign or malignant [69].

Moreover, the huge amount of blood vessels and the autonomic innervation which regulate their action, can help to understand how alterations in thermoregulation, lymphatic flow, and venous circulation are related to the SF. Caggiati et al. (1999) highlighted how the SF represents an excellent support for the wall of the saphenous veins, supporting the adventitia from the outside through numerous fibrous septa which act like the spokes of a bicycle [70] (Figure 2A). Consequently, if the SF is altered, it can stretch the venous wall in an anomalous way, causing a deformation of the vein and consequently incontinence of the venous valves (Figure 2B).

The recent demonstration of the huge sympathetic innervation of the SF suggests that it could also be influenced by a state of chronic stress [71,72]. The changes caused by chronic stress can also influence the immune system through the axis TGF-β1/Smad2/3/Foxp3, causing lymphocyte apoptosis and immunosuppression [73]. These aspects suggest a possible involvement of the SF also in the mechanism of fibromyalgia, in which patients show a reduced dermal fiber length of fibers with vessel contact, suggesting a possible relationship between sympathetic neurons and impaired thermal tolerance commonly reported by fibromyalgic patients [74].

## 3. Materials and Methods

Our search was conducted on Web of Science (WOS), PubMed, and Cochrane databases, covering publications from their inception until November 2024. The MeSH keyword employed was “Superficial fascia”, following the string PubMed: (superficial fascia [Title/Abstract]); WOS: (superficial fascia [title]); Cochrane: (superficial fascia [Title Abstract Keyword]). The inclusion criteria were studies about superficial fascia in humans, in English language. The exclusion criteria were studies related in general to ‘fasciae’ but referred to the deep one without specifying it; those not written in English; studies on animals; and duplicates.

The literature search was conducted by one reviewer (C.C) and verified by a senior researcher (C.F.). Any discrepancies were resolved through consensus among the authors. Our screening process involved reviewing titles and abstracts, followed by a full-text review of eligible studies. Additionally, we meticulously examined the references of the included studies to identify any further relevant publications. In total, 1056 papers were identified (PubMed: 582, WOS: 147, Cochrane: 327). After removing 23 articles as duplicates and 96 as ineligible by automation tools, the screening was conducted according to the inclusion and exclusion criteria, removing papers not in English (95) and papers not related to fascia (622). Subsequently, the textual content of the remaining 220 potentially eligible papers was meticulously reviewed, removing papers that did not align with our predefined inclusion criteria. Ultimately, 60 articles remained available, and 4 more papers were added by hand search to identify other potentially eligible studies for inclusion in the analysis. Finally, 64 studies met the inclusion criteria and were considered eligible for the comparative analysis for the review (Figure 3).

## 4. Conclusions

The SF is recognized as a specific anatomical structure with distinctive cellular, innervation, and vascularization properties. It is a thin, fibrous membrane in the hypodermis that extends continuously throughout the body and is composed of irregularly arranged collagen fibers interspersed with numerous elastic fibers. It works as a scaffold for supporting adipose lobules, providing structural integrity to the surrounding tissues, and guaranteeing autonomy between the skin and the muscle/deep fascia. However, the more the SF is studied, the more it is discovered that it encompasses multiple functions. It is implicated in tissue assemblage and mobilization, opening new perspectives for wound repair. It can be involved in reconstructive surgery and pain management of clinical manifestations caused by altered lymphatic transport (like lymphedema). Lastly, the rich thin autonomic innervation supplying the tissue can be influenced by a stress condition or a sudden change in temperature.

All this evidence highlights the complex and fascinating role of the SF that should be considered in clinical practice. Indeed, a better understanding of its dynamics will help in a good comprehension of some fascial dysfunctions.

## Figures and Tables

**Figure 1 ijms-26-01289-f001:**
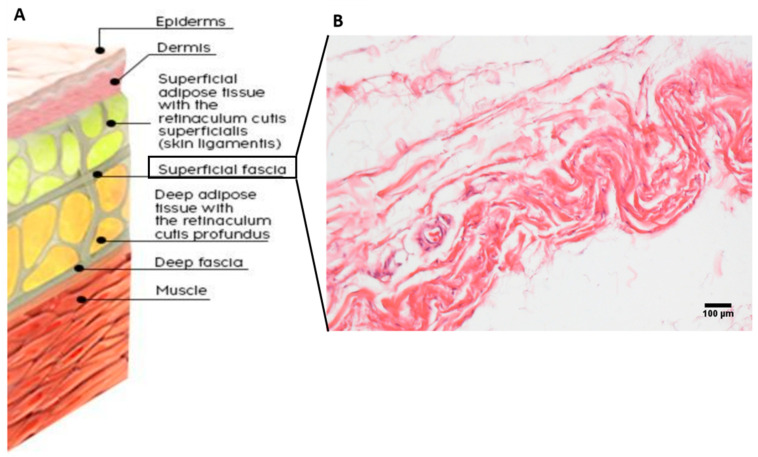
Scheme representing the anatomy from skin to the muscle (**A**) and histological detail (Hematoxylin and Eosin staining) of the SF layer (**B**).

**Figure 2 ijms-26-01289-f002:**
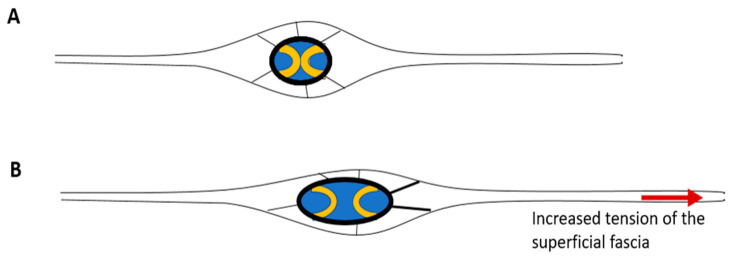
Schematic representation of the saphenous vein (in blue) with its valves (in yellow) and its relationship with the SF, depicted around the vein (in black and white), in physiological (**A**) and anomalous (**B**) condition. (**A**) as described by Caggiati [70], the SF splits around the saphenous vein and sends some fibrous septa to the adventitia, supporting from outside the patent of the vein. (**B**) when SF is overstretched due to a scar or fascial fibrosis (red arrow), the anomalous tension is transmitted to the support system of the saphenous vein, causing a deformation of the vessel and consequently incontinence of the valves (in yellow).

**Figure 3 ijms-26-01289-f003:**
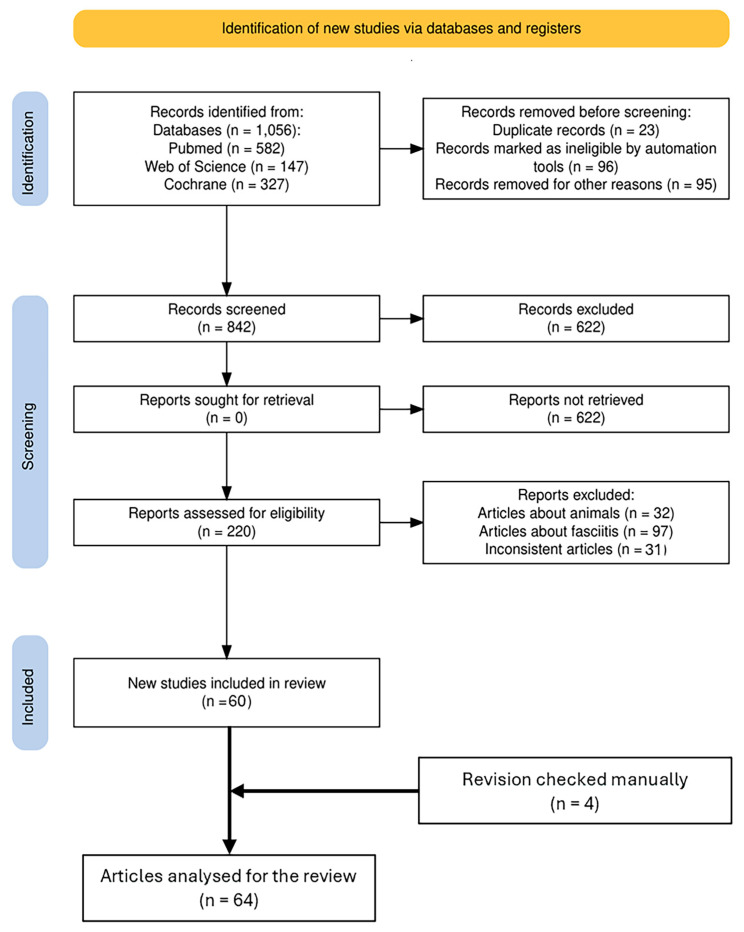
PRISMA flow diagram of articles selection. From: Page MJ, McKenzie JE, Bossuyt PM, Boutron I, Hoffmann TC, Mulrow CD, et al. The PRISMA 2020 statement: an updated guideline for reporting systematic reviews [75].

**Table 1 ijms-26-01289-t001:** Values of SF thicknesses (in µm,) measured using US imaging.

Region		Mean Value	Reference
Upper limb	Anterior	400 ± 100	Pirri et al., 2022 [27]
	Posterior	530 ± 100	
Thorax		560 ± 120	Hammoudeh et al., 2022 [26]
Dorsal Trunk	Cranial	600 to 700	Hammoudeh et al., 2022 [26]
	Caudal	500 to 600	
Abdomen	Cranial	364 ± 22 ♂ 315 ± 56 ♀	Kumar et al., 2011 [30] Hammoudeh et al., 2022 [26]
	Caudal	528 ± 38 ♂ 390 ± 36 ♀	
Lumbar region		580 ± 100	Hammoudeh et al., 2022 [26]
NAC		309 ± 171	Matousek et al., 2014 [31]
Thigh	Anterior	490 ± 140	Pirri et al., 2022 [25]
	Posterior	500 ± 110	
	Medially	520 ± 100	
	Laterally	420 ± 120	

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
