# Peer review of "The Human Superficial Fascia: A Narrative Review"

_ijms, 2025, doi:10.3390/ijms26031289_

Round 1

Reviewer 1 Report

Comments and Suggestions for Authors

Article: The superficial fascia: a narrative review

Authors: Caterina Fede, Claudia Clair, Carmelo Pirri, Lucia Petrelli, Xiao Xiao Zhao, Yunfeng Sun, Veronica Macchi and Carla Stecco

This is an interesting and detailed Review about the superficial fascia. The methodology is correct and the relevance of the topic reviewed make this paper worthy of publication. However, there are some minor concerns to the paper that should be addressed in the interest of clarity, and in order to improve the quality of the manuscript. Specific comments are outlined, point by point, below.

1- Lines 24 and 25: "...superficial fascia..." should be replaced by "...superficial fascia (SF)...". Please see line 101 of this manuscript. Please check all the manuscript and either replace "superficial fascia" by "SF" or remove all the abbreviations "SF".  Please standardize.

2- Lines 24-26: “However, the superficial fascia in recent years was recognized as a key element that organize the hypodermis. It is a thin fibrous layer mainly formed by collagen and elastic fibers, but also, containing fat lobules, vessels and nerves.” Please add references to these sentences.

3- Lines 25-26: “It is a thin fibrous layer mainly formed by collagen and elastic fibers, but also, containing fat lobules, vessels and nerves.” Actually, it also contains muscle fibers in some body regions. Please see lines 141-146 of this manuscript (please see also, for instance, Gray’s Anatomy. The Anatomical Basis of Clinical Practice. Elsevier, 42nd ed., pag. 152). In my opinion, the mention to the presence of muscle fibers in the superficial fascia in some body regions should be included in the aforementioned sentence.

4- Lines 62-64: Concerning the face, in the Reviewer's opinion, the buccinator muscle must be specified here regarding its relation with the buccopharyngeal fascia. (please see, for instance, Gray’s Anatomy. The Anatomical Basis of Clinical Practice. Elsevier, 42nd ed., pag. 625).

5- Lines 96: “...Us images...” should be replaced by “...ultrasound (US) images...”.

6- Lines 112-114: In the Reviewer's opinion, the nomenclature of trunk regions used here in the text and those used in the Table 1 should be similar to avoid ambiguities.

7- Line 126: In the Reviewer's opinion, the Table 1 would be more complete if it included a column with the bibliographic references were the specific data were taken. Although these references are in the text, the Reviewer think that repetition would be very useful in this case.

8- Lines 155 and 156: References should be added to this sentence.

9- Line 176: “…strength and resultant Young's modulus…”. Please check.

10- Line 285: “…greater flexion of the skin…”. In the Reviewer opinion, another way must be found to describe precisely what happens with the skin in the lower part of the anterior abdominal wall.

11- Lines 332: In order to be accurate,"...saphenous vein..." should be replaced by "...long or short saphenous vein..."

12- Figure 2. In the Reviewer opinion, the reasons that underlie the remotion of the 95 records should be briefly mentioned in text of Material and methods before the Figure.

In conclusion, the topics mentioned above should be addressed if the goal of this Review is to clarify the important questions that it aims to address. The Reviewer think that the authors would gain from taking note of the comments and/or suggestions made here and of the possible comments and suggestions made by other Reviewers. Finally, the Reviewer apologize the Editor and/or the Authors if she/he have unintentionally missed any important issue and/or if any of the questions she/he raise do not make sense and/or are not particularly relevant. Thank you very much.

Author Response

This is an interesting and detailed Review about the superficial fascia. The methodology is correct and the relevance of the topic reviewed make this paper worthy of publication. However, there are some minor concerns to the paper that should be addressed in the interest of clarity, and in order to improve the quality of the manuscript. Specific comments are outlined, point by point, below.

Thanks to the reviewer for the valuable time and effort in reviewing our work, and for suggesting improvements, which we have explained and implemented below.

1- Lines 24 and 25: "...superficial fascia..." should be replaced by "...superficial fascia (SF)...". Please see line 101 of this manuscript. Please check all the manuscript and either replace "superficial fascia" by "SF" or remove all the abbreviations "SF".  Please standardize.

Thank you for the right suggestion; we have corrected the entire text by inserting the abbreviation SF.

2- Lines 24-26: “However, the superficial fascia in recent years was recognized as a key element that organize the hypodermis. It is a thin fibrous layer mainly formed by collagen and elastic fibers, but also, containing fat lobules, vessels and nerves.” Please add references to these sentences.

Thank you, we have added 3 references to the sentence.

3- Lines 25-26: “It is a thin fibrous layer mainly formed by collagen and elastic fibers, but also, containing fat lobules, vessels and nerves.” Actually, it also contains muscle fibers in some body regions. Please see lines 141-146 of this manuscript (please see also, for instance, Gray’s Anatomy. The Anatomical Basis of Clinical Practice. Elsevier, 42nd ed., pag. 152). In my opinion, the mention to the presence of muscle fibers in the superficial fascia in some body regions should be included in the aforementioned sentence.

Yes, we agree with the reviewer. Although we have described this aspect further on in the text (Paragraph 2.2. The cellular population), we have here modified the description of the SF by adding the fact that in some regions it also presents muscle cells: “It is a thin fibrous layer mainly formed by collagen and elastic fibers, but also, containing fat lobules, vessels and nerves [5-7], and in some regions including also muscle fibers [6].”

4- Lines 62-64: Concerning the face, in the Reviewer's opinion, the buccinator muscle must be specified here regarding its relation with the buccopharyngeal fascia. (please see, for instance, Gray’s Anatomy. The Anatomical Basis of Clinical Practice. Elsevier, 42nd ed., pag. 625).

Revised, thank you. The new sentence is: “Even at the facial level, points of adhesion between the superficial and deep fascia have been identified, over either the masseter or buccinator muscles, with the function of separating adjacent adipose compartments, or adipose compartments and anatomical spaces [24].”

5- Lines 96: “...Us images...” should be replaced by “...ultrasound (US) images...”.

Done, thank you.

6- Lines 112-114: In the Reviewer's opinion, the nomenclature of trunk regions used here in the text and those used in the Table 1 should be similar to avoid ambiguities.

Good point, thank you for highlighting it.

We have modified both the text and the Table. The new text is: “The dorsal trunk has the thickest SF with a mean thickness of 600 µm (0.6 to 0.7 mm), followed by the lumbar region (similar thickness, but slightly thinner on average, ~580 µm) [26].

The anterior regions appear thinner, with an average SF thickness of 560 µm in the thorax [26].”

Then, we have added some lines in the Table, with all the new values of mean thickness, for the anterior region (thorax and abdomen), and for the posterior region (dorsal trunk and lumbar region).

7- Line 126: In the Reviewer's opinion, the Table 1 would be more complete if it included a column with the bibliographic references were the specific data were taken. Although these references are in the text, the Reviewer think that repetition would be very useful in this case.

We agree with the good suggestion, we have added a column with the references.

8- Lines 155 and 156: References should be added to this sentence.

Added, thank you.

9- Line 176: “…strength and resultant Young's modulus…”. Please check.

It's a typo. Corrected.

10- Line 285: “…greater flexion of the skin…”. In the Reviewer opinion, another way must be found to describe precisely what happens with the skin in the lower part of the anterior abdominal wall.

We have modified with “sagging fold of the skin”, that is the term used by the Authors of the cited reference. Thank you for the indication.

11- Lines 332: In order to be accurate,"...saphenous vein..." should be replaced by "...long or short saphenous vein..."

In our opinion, it is not appropriate “short or long”, but we appreciate the suggestion and we have revised the legend to make it clearer: “Schematic representation of the saphenous vein (in blue) with its valves (in yellow) and its relationship with the SF, depicted around the vein (in black and white), in a physiological (A) and anomalous (B) condition.”

12- Figure 2. In the Reviewer opinion, the reasons that underlie the remotion of the 95 records should be briefly mentioned in text of Material and methods before the Figure.

Done, thank you. The new sentence in Material and Methods is: “After removing 23 articles as duplicates and 96 ineligible by automation tools, the screening was conducted according to the inclusion and exclusion criteria, removing papers not in English (95) and papers not related to fascia (622).”

In conclusion, the topics mentioned above should be addressed if the goal of this Review is to clarify the important questions that it aims to address. The Reviewer think that the authors would gain from taking note of the comments and/or suggestions made here and of the possible comments and suggestions made by other Reviewers. Finally, the Reviewer apologize the Editor and/or the Authors if she/he have unintentionally missed any important issue and/or if any of the questions she/he raise do not make sense and/or are not particularly relevant. Thank you very much.

Thank you again for the encouraging and insightful comments. We edited our manuscript carefully following suggestions to improve the manuscript.

Reviewer 2 Report

Comments and Suggestions for Authors The manuscript by Fede et al. performs an exhaustive literature review on the superficial fascia in humans. Although the topic is interesting and there is great value in gathering the little knowledge on the subject, I believe that it should be completely restructured for publication. It is not understood why if it is a bibliographic review it follows the format of an original work. I think it is not worth including a results section in this type of work. The clarity of the explanation regarding the selection of the references used stands out. I think the title should be clarified that the review is about the superficial fascia in humans, because, despite what the authors mention and should correct, the superficial and deep fascia are found in other mammals. In several paragraphs, the wording becomes confusing and for example on Page 1Line 40-42 as it is written it seems that they consider the hypodermis part of the skin or on Page2 line 75, the fibers are extracellular matrix, please change the sentence The creation of some tables summarizing the characteristics by area could favor the understanding of the work. Other aspects: Line 142 What animals are you referring to when you mention the fleshy panicle? Please be more specific from a taxonomic point of view. Finally, the work almost does not review molecular aspects, I think it could be more suitable for another journal.

Author Response

Comments and Suggestions for Authors

The manuscript by Fede et al. performs an exhaustive literature review on the superficial fascia in humans. Although the topic is interesting and there is great value in gathering the little knowledge on the subject, I believe that it should be completely restructured for publication.

Authors thank to the reviewer for his valuable time and effort in reviewing our work, and for the comments and suggestions. We have improved the work as indicated by all three Reviewers, and we hope that the new version will be suitable for publication and of interest to readers.

It is not understood why if it is a bibliographic review it follows the format of an original work.

The instruction for Authors of IJMS indicate that “Structured reviews and meta-analyses should use the same structure as research articles and should ensure they conform to the PRISMA guidelines.”

For this reason, we have accurately followed the indicated format.

I think it is not worth including a results section in this type of work. The clarity of the explanation regarding the selection of the references used stands out.

As explained above, we followed the journal’s instructions, both for methods and selection of the references and for the exploration of the topic. Thank you for recognizing our effort in explaining this topic still little explored and known.

I think the title should be clarified that the review is about the superficial fascia in humans, because, despite what the authors mention and should correct, the superficial and deep fascia are found in other mammals.

We agree. Thank you for the right suggestion, in this review we were specifically focused on describing the human anatomy and physiology of the superficial fascia. The new title of the Review is: “The human superficial fascia: a narrative review”. We added “human” also in the Abstract.

In several paragraphs, the wording becomes confusing and for example on Page 1 Line 40-42 as it is written it seems that they consider the hypodermis part of the skin.

The two layers we are referring to are the SAT and the DAT, we have now made it explicit: “Passing between these two layers, the SAT and the DAT, and dividing them, the SF permits to separate the skin form the musculoskeletal system, allowing an independent and normal sliding of these two components.”  

or on Page2 line 75, the fibers are extracellular matrix, please change the sentence.

We agree, thank you for the observation. Now it is revised in “The SF presents the typical characteristics of a fibrous connective tissue formed by cells, fibrous and aqueous extracellular matrix”.

The creation of some tables summarizing the characteristics by area could favor the understanding of the work.

We have improved the Table of the Review, with the mean thicknesses of the main anatomical sites, adding also the References. We decided to not add any other tables by area because the review is narrative, and it considers the different characteristics of the fibrous component and cellular component of the SF in general, in humans, and not a study per area and/or anatomic site. Surely, in the next future it could be a good suggestion for a second work related to differentiate the SF per area.

Other aspects: Line 142 What animals are you referring to when you mention the fleshy panicle? Please be more specific from a taxonomic point of view.

Thank you, we refer to lower mammals, we have revised the sentence: “Moreover, the SF, having ancestral origin from the panniculus carnosus present in lower mammals, which serves to produce local movement of the skin [16], can present in some regions also muscle fibers,[…]”.

Finally, the work almost does not review molecular aspects, I think it could be more suitable for another journal.

It is a right observation, but we have already published a similar review about the deep/muscular fascia in the same journal in 2021 (Fede C, Pirri C, Fan C, Petrelli L, Guidolin D, De Caro R, Stecco C. A Closer Look at the Cellular and Molecular Components of the Deep/Muscular Fasciae. Int J Mol Sci. 2021 Jan 30;22(3):1411). We think that for continuity it would be interesting to publish a second review related to the SF with a similar structure in the same journal.

Reviewer 3 Report

Comments and Suggestions for Authors

I would like to compliment you on how you have explored this topic that is certainly not commonly known. I appreciated both the anatomical description and the physiology arguments. I expect that in the future, not too distance, we can discuss the implications that the fascia has on the pathologies of the blood-lymphatic system and on the disorders of the fat organ.

The article aims to describe the superficial Fascia (and the deep one) which is a structure little considered by health professionals as it is considered a passive containment structure in the physiology and physiopathology of the human body. This article instead emphasizes its active aspect, where the fascia participates both in the differentiation of the tissues that are adjacent to it and in the regulation of the vital fluids that are distributed under the skin.
The contents that it presents make it sufficiently original but certainly very pertinent to the initial objective. In fact, the regulation of blood and subcutaneous lymph fluid may be the basis of some pathologies that are little studied today. Further research could develop, in an original way compared to how it is done today, a study of the physiopathology of primary and secondary Lymphedema. In this it may be useful to investigate, within the range, the changes that occur during the growth of the child/man capable of inducing the manifestation of primary lymphedema which in 90% of cases occurs in the first decade of life or in the 3rd decade.
Remaining with the evaluation of the article I recognize that it is based on a methodology suitable for having appropriate and coherent data with the anatomical and physiological truth of the region under examination. Hence the conclusions adequate to the starting objectives.
The bibliography that accompanies it is very extensive. This demonstrates a good relationship between the authors and the scientific world and with researchers who are interested in this topic. The synergy between researchers always leads to new and important results.

Author Response

I would like to compliment you on how you have explored this topic that is certainly not commonly known. I appreciated both the anatomical description and the physiology arguments.

We thank the Reviewer for appreciating our work and for the time and effort dedicated to its review.

I expect that in the future, not too distance, we can discuss the implications that the fascia has on the pathologies of the blood-lymphatic system and on the disorders of the fat organ.

We agree, and we are studying the role of superficial fascia in pathologies like lymphedema and lipedema.

The article aims to describe the superficial Fascia (and the deep one) which is a structure little considered by health professionals as it is considered a passive containment structure in the physiology and physiopathology of the human body. This article instead emphasizes its active aspect, where the fascia participates both in the differentiation of the tissues that are adjacent to it and in the regulation of the vital fluids that are distributed under the skin.
The contents that it presents make it sufficiently original but certainly very pertinent to the initial objective. In fact, the regulation of blood and subcutaneous lymph fluid may be the basis of some pathologies that are little studied today. Further research could develop, in an original way compared to how it is done today, a study of the physiopathology of primary and secondary Lymphedema. In this it may be useful to investigate, within the range, the changes that occur during the growth of the child/man capable of inducing the manifestation of primary lymphedema which in 90% of cases occurs in the first decade of life or in the 3rd decade.

We absolutely agree, and our team is doing a huge effort in understanding the role of connective tissue in lymphedema, and in visualizing a potential anatomical predisposition, to improve the early diagnosis and the management of the pathology (see papers by Pirri and coauthors).

Remaining with the evaluation of the article I recognize that it is based on a methodology suitable for having appropriate and coherent data with the anatomical and physiological truth of the region under examination. Hence the conclusions adequate to the starting objectives.
The bibliography that accompanies it is very extensive. This demonstrates a good relationship between the authors and the scientific world and with researchers who are interested in this topic. The synergy between researchers always leads to new and important results.

Thank you again, we appreciate these positive comments. We are sure that with the advances in the research about fasciae we can understand better their physiology and how they change in the pathology, with the aim to improve the personalized medicine and a better targeted therapy for patients.

Round 2

Reviewer 2 Report

Comments and Suggestions for Authors The authors have made the suggested corrections and satisfactorily answered the questions asked. The work is ready to be published, only lower mammals should be changed to non-human mammals, in current biology it is not correct to use the terms superior and inferior to refer to different species.

Author Response

The authors have made the suggested corrections and satisfactorily answered the questions asked. The work is ready to be published, only lower mammals should be changed to non-human mammals, in current biology it is not correct to use the terms superior and inferior to refer to different species.

Thank you, we have corrected in "non-human".